# Lumbar Temperature Map of Elderly Individuals with Chronic Low Back Pain—An Infrared Thermographic Analysis

**DOI:** 10.3390/diagnostics15111317

**Published:** 2025-05-23

**Authors:** Nelson Albuquerque, Liliana Gonçalves, Wally Strasse, Joaquim Gabriel, Laetitia Teixeira, Pedro Cantista

**Affiliations:** 1Abel Salazar Institute of Biomedical Sciences, University of Porto, 4050-313 Porto, Portugal; 2Termalistur, Termas de São Pedro do Sul, 3660-692 Várzea, Portugal; 3Federal University of Technology, Curitiba 81.030-001, Paraná, Brazil; 4Faculty of Engineering, University of Porto, 4099-002 Porto, Portugal

**Keywords:** infrared thermography, low back pain, elderly

## Abstract

**Background/Objectives**: Chronic low back pain (CLBP) is a prevalent condition that significantly impacts the aging population. Among non-invasive assessment tools, infrared thermography (IRT) has been highlighted as a radiation-free method to evaluate thermal variations in the lumbar region. However, its applicability in clinical practice and correlation with functional and pain-related parameters remain unclear. This study aimed to analyze the thermal profile of the lumbar region in elderly individuals with CLBP and explore potential correlations between lumbar temperature patterns and clinical factors such as pain intensity and functional capacity. **Methods**: A cross-sectional observational study was performed in an outpatient setting. The population included thirty-one elderly individuals diagnosed with CLBP. IRT was used to assess the lumbar temperature distribution, including participants who reported pain radiating to the lower limbs. Pain intensity was measured using a numerical rating scale (0–10). The functional assessments included spine mobility tests and validated questionnaires evaluating clinical characteristics. **Results**: No significant differences in lumbar temperature patterns were observed among the participants. Additionally, no correlation was found between pain intensity and functional capacity based on a thermographic analysis. Nonetheless, individuals reporting lower fatigue levels and those with a higher body mass index (BMI) were generally associated with cooler thermal readings on the lumbar region’s thermographic maps. **Conclusions**: These findings suggest that IRT may require methodological refinements, including optimized technical specifications and image acquisition protocols, to enhance its applicability in assessing CLBP. Indeed, IRT might not be the most effective tool for evaluating pain-related thermal changes in elderly populations. Further research is needed to clarify its role in clinical assessments.

## 1. Introduction

Chronic low back pain (CLBP) has been extensively recognized as a prevalent health issue, with foundational research by scholars [1,2] underscoring its prevalence. The condition has emerged as the foremost reason for diminished activity and work absenteeism across the globe [3]. The socioeconomic impact of CLBP is profound, straining not only the affected individuals but also their families, communities, and burdened economic structures [4,5,6].

Defined by persistent pain in the lumbar area without a clear anatomopathological cause, as elucidated in studies [7,8,9], CLBP poses a complex diagnostic challenge [10]. A holistic evaluation encompassing biomechanical, sensory, and autonomic dimensions could offer deeper insights into the condition. Hence, infrared thermography (IRT) [11] presents a straightforward method for investigating autonomic irregularities.

IRT has been employed in the domain of musculoskeletal disorders for decades, such as rheumatic diseases, for example, tracing back to the pioneering work in the thermographic evaluation of pain [12,13]. Subsequent research has both supported and contested its efficacy, with notable findings demonstrating IRT’s sensitivity, its correlation with other diagnostic tools (e.g., CT scans, myelography, and MRI), and its ability to detect abnormalities linked to lumbar radiculopathy [13,14].

Clinically, spinal pathologies and/or pain often manifest as myofascial trigger points, muscle fatigue, and soreness, particularly in the paravertebral muscles. Static infrared thermography has been employed, for example, to assess the temperature distribution in scoliosis patients, aiding in the evaluation of muscular imbalances [15]. Yet, standard imaging tests, such as X-rays, CT scans, or MRIs, frequently fail to reveal abnormalities in patients who continue to experience pain [16]. This discrepancy underscores the potential of IRT to uncover such clinical signs, bolstering arguments for its utility [17]. IRT’s capacity to detect changes in body temperature occurring due to multiple factors—ranging from physical exertion and metabolic shifts to rheumatic and musculoskeletal conditions—underscores its diagnostic value. This temperature variation serves as an indicator of underlying physiological processes. The technique’s non-contact, non-invasive nature, coupled with its rapidity and the ease of interpreting color-coded images, has facilitated its adoption in clinical settings. Importantly, IRT is safe, leveraging the natural radiation emitted by the skin, and its affordability and simplicity make it suitable for regular use. Therefore, recent developments in device technology and computational methods have further elevated interest in IRT as a diagnostic tool [18]. However, some researchers remain skeptical about its application for CLBP due to perceived limitations in its diagnostic accuracy, stability, and specificity [19,20]. Controversy persists, supported by studies reporting divergent findings on skin temperature variations in CLBP patients [21,22].

Beyond its diagnostic applications, IRT has been advocated as a valuable tool for monitoring treatment efficacy, being capable of identifying deviations from normative physiological states [23]. Recent studies have demonstrated that IRT may serve as a sensitive, non-invasive method for identifying lumbosacral radicular pain, especially when clinical signs are inconclusive [24].

This study aimed to evaluate autonomic nervous system activity through thermographic assessments, providing insights into thermal profiles that may correlate with the condition. Pain intensity, functionality, fatigue, strength, and other physical, sociodemographic, anthropometric, and clinical data were collected to provide a broader context for understanding the condition. Ultimately, this study sought to identify potential patterns and contributing factors that could inform future strategies for the treatment and rehabilitation of elderly individuals with CLBP. In summary, this study addresses a critical gap in non-invasive diagnostic tools for CLBP in the elderly by exploring IRT, a radiation-free technique that aligns with current trends toward safer and more patient-friendly diagnostic approaches.

## 2. Materials and Methods

### 2.1. Study Design

In this study, a cohort of 31 individuals suffering from CLBP, with some experiencing radiating pain to the lower extremities, was examined. The participant group comprised 20 men and 11 women, whose ages ranged from 66 to 88 years. Before the commencement of this study, each participant was thoroughly briefed on the study’s procedures, potential risks, and benefits. They provided written informed consent, agreeing to participate in the study and allowing their data to be utilized for research purposes. This consent process ensured that all participants were fully informed and agreed to their involvement in the research.

The study was submitted to the joint Ethics Committee of Instituto de Ciências Biomédicas de Abel Salazar of Porto University (ICBAS-UP) and was reviewed in a plenary meeting held on 1 May 2022, receiving a favorable opinion on that date. The approval was formalized under reference number SCE/HCC/010, issued on 22 July 2022, corresponding to protocol 2021/CE/P31/(P371/CETI). The research was carried out in strict adherence to the ethical guidelines outlined in the Declaration of Helsinki, ensuring the ethical integrity of the study.

The participants were selected based on clearly defined inclusion and exclusion criteria. The inclusion criteria consisted of being 65 years of age or older, presenting with CLBP (with or without radiation to the lower limbs) persisting for more than three months, and having lumbar spine imaging performed within the past five years that excluded neoplastic conditions or significant structural misalignments. The exclusion criteria included receiving balneotherapy or physical and rehabilitation medicine treatments within the previous three months, illiteracy, refusal to abstain from tobacco and caffeine within 12 h before each thermographic assessment, a history of spinal surgery, prior spinal fracture, infection, or neoplasia, and moderate-to-severe scoliosis. The additional exclusion criteria encompassed neuromuscular deficits in the lower limbs, bladder or bowel dysfunction, presence of fever or chills, recent major trauma, unexplained weight loss, night sweats, or persistent nocturnal pain. Individuals diagnosed with active systemic autoimmune diseases (e.g., rheumatoid arthritis, systemic lupus erythematosus, systemic sclerosis, Sjögren’s syndrome, antiphospholipid antibody syndrome, polymyositis, or dermatomyositis) and those who had used anti-inflammatory drugs or corticosteroids within the past 30 days were also excluded.

To ensure the accuracy and reliability of the thermographic evaluations, this study adhered to a standardized operating procedure. These evaluations were critical in assessing the thermal profile of each participant, which is indicative of autonomic nervous system activity and could provide insights into their condition. The level of pain experienced by the participants was quantitatively measured using a numerical pain scale that ranged from 0 to 10, allowing for a standardized assessment of pain intensity. Pain was assessed using a unidimensional numeric scale (0–10), which, while practical, did not capture the multidimensional nature of chronic pain. Future studies may benefit from incorporating instruments like the McGill Pain Questionnaire to enrich the clinical characterization.

In addition to the thermographic and pain assessments, this study utilized various questionnaires to gather comprehensive sociodemographic, anthropometric, and clinical data from the participants. Among the tools employed were the Oswestry Disability Index, version 2.0, which is widely used to gauge the degree of disability related to low back pain, and the Fatigue Assessment Scale, which measures fatigue levels. These instruments helped in understanding the broader impact of CLBP on the participants’ quality of life and functional abilities.

Moreover, the participants underwent a battery of spinal mobility tests, which included the Shober test, finger-to-floor distance, occiput-to-wall distance, and lateral flexion assessments. These tests provided quantitative and qualitative data on the spinal mobility and flexibility of the participants. Manual grip strength assessments were also conducted to evaluate the muscular strength and function of the participants. This comprehensive evaluation approach allowed for a multifaceted understanding of the impact of CLBP on the participants, encompassing pain levels, functional disability, fatigue, and spinal mobility.

### 2.2. Instruments

To prepare for the thermographic evaluations, the study participants were advised to remove their clothing to facilitate a direct assessment of the skin in the lumbar region. This step was essential to ensure the accuracy of the thermographic images. To enhance methodological rigor, standardized procedures and validated instruments were employed, with particular attention to environmental controls, including room temperature stabilization, thermoequilibration periods, and fixed camera positioning, to ensure technical precision. They were then allowed a thermoequilibration period of 10–15 min in a controlled environment, where the room temperature was maintained at 24 °C (with a standard deviation of 2.77 °C). This process ensured that the participants’ body temperatures stabilized, minimizing external influences on the skin temperature. After this preparatory phase, thermographic images of the participants’ back area were captured while they were seated comfortably on a chair. For a precise analysis, the lumbar region of interest was marked out using a polygonal outline that spanned from the first to the fifth lumbar vertebra, as depicted in Figure 1 of the study documentation.

The thermographic data acquisition was carried out using an FLIR I7^®^ infrared camera, equipped with a detector resolution of 140 × 140 pixels and a thermal sensitivity of less than 0.10 °C. Although the FLIR i7 camera has a lower resolution compared to other clinical thermography models, its selection was based on accessibility, portability, and prior validation in similar studies. To optimize the accuracy of the thermal images, the camera’s emissivity level was adjusted to 0.98. The camera was strategically positioned 0.8 m away from the subjects in a room where both the temperature and humidity were monitored to ensure consistent and reliable measurements.

The analysis of the thermographic data was conducted using FLIR Research Studio R&D software, version 3.2.2. During this analysis phase, all captured images were standardized to a uniform scale, and seven specific anatomical points on the lumbar region were selected for detailed examination, following a pre-established protocol, as outlined in Figure 1. This meticulous approach allowed for the consistent and precise measurement of thermal variations across the defined points of interest. The seven selected thermographic analysis points were chosen. The ROIs 1 and 2 were delimited 1 cm below the T12 vertebra, ending 1 cm below the waistline of the body, corresponding to a perpendicular line dashed horizontally to the end of the epicondyle bone line (located on the external lateral line of the elbow) and 2 cm laterally distant from the spine, in the evaluation of the temperature of the medial portion of the serratus muscle. ROIs 2 and 3 were delimited on the dashed line perpendicular to the L4 vertebra and evaluated 3 cm distant from the spine. ROI 5 was delimited 1 cm below L5, at the central point of the piriformis muscle. ROIs 6 and 7 were delimited using the reference of the beginning of the gluteal fissure, located 3 cm below the central point of the gluteus maximus muscle, and evaluated 4 cm laterally from the spine. All ROIs have 9 × 9 pixels.

For the statistical analysis of the collected data, the study employed the SPSS statistical software (Version 26; Statistical Package for the Social Sciences), a comprehensive tool for the analysis of social science data. The statistical methods applied in the analysis included the Mann–Whitney test, Spearman’s correlation, and linear regression analyses, allowing for the exploration of relationships between thermographic findings and clinical variables. A significance level of *p* < 0.05 was established, meaning that observed differences or correlations with a probability value less than 0.05 were considered statistically significant, indicating a meaningful deviation from chance. This rigorous analytical approach provided a solid foundation for interpreting the thermographic data in the context of CLBP and its clinical manifestations. The use of non-parametric tests, such as the Mann–Whitney U-test and Spearman’s correlation, was justified by the distribution of the data, enabling a robust assessment of the relationships between thermal profiles and clinical parameters like pain intensity, fatigue, spinal mobility, and body mass index (BMI).

### 2.3. Data Analysis

Our study adopted a comprehensive approach to analyze the data, using a variety of statistical parameters, like frequency, percentage, mean, and standard deviation, to map the distribution and characteristics of our variables. This initial analysis helped outline the data’s distribution and central tendencies, setting the stage for a deeper investigation. Considering our relatively small sample size, we considered non-parametric tests over parametric ones for inferential analysis, based on the Kolmogorov–Smirnov normality test results indicating non-normal distribution in our data. This led us to choose the Mann–Whitney test for comparing demographic and clinical variables, such as gender, age, pain type, morning stiffness, and low back pain or sciatica episodes, due to its appropriateness for non-normally distributed data, allowing for a robust comparison between groups. Additionally, we used Spearman’s correlation coefficient to explore the strength and direction of associations between variables, given its suitability for ranked, non-parametric data. The findings are presented through tables and graphs for clarity and ease of interpretation, with each accompanied by a thorough analysis to enhance understanding. This methodological detail and transparent presentation underscore the depth and seriousness of our investigation into the complexities of CLBP.

## 3. Results

In our investigation, we analyzed the thermographic temperatures in the lower back of elderly individuals with CLBP, assessing potential links between lumbar temperatures and various demographic and clinical factors, including gender, age, pain characteristics, morning stiffness, and episodes of acute pain or sciatica. Using the Mann–Whitney test to compare groups, we particularly focused on gender differences in lumbar temperatures across seven specific points (P1 to P7). Despite a thorough analysis, as documented in Table 1, we found no significant temperature differences between male and female participants, indicating that gender does not significantly impact the lumbar thermal profile in CLBP sufferers. This conclusion enriches our understanding of CLBP, suggesting similar thermal patterns across genders and highlighting the need for further research into CLBP’s physiological aspects and management in the elderly.

In our continued exploration, we investigated whether age impacts the thermographic temperatures in the lumbar region among CLBP sufferers, hypothesizing that age-related physiological differences might influence the thermal profile of the lower back. We meticulously analyzed thermographic data from seven lumbar points (P1 to P7), categorizing our participants by age to identify any temperature patterns that could be associated with aging.

The analysis, as detailed in Table 2, aimed to uncover how aging might affect the thermal signatures linked to CLBP. However, our findings revealed no statistically significant temperature differences across age groups, indicating that within our study’s context, age did not significantly influence the lumbar region’s thermal profiles in CLBP patients. This challenges prior assumptions regarding age’s role in CLBP’s pathophysiology, suggesting consistent thermal characteristics across elderly age brackets. The lack of significant age-related thermal variations highlights the complexity of CLBP and the importance of broader factors in its diagnosis and management, laying the groundwork for future research into the condition’s multifaceted nature.

Expanding our analysis, we investigated how the thermographic temperature profiles in the lumbar region correlate with pain characteristics in CLBP sufferers. We aimed to determine if the pain’s nature—localized in the lumbar region or radiating to the lower limbs—affected the lumbar area’s thermal signatures. This inquiry was based on the hypothesis that the way pain manifests could influence the thermal patterns observed, potentially revealing the mechanisms behind pain distribution. We analyzed thermographic data from specific lumbar points (P1 to P7), categorizing participants by their reported pain type. The findings detailed in Table 3 showed no significant temperature differences between those with localized versus radiating pain. This indicates that, within our study’s context, the pain’s manifestation did not significantly alter the lumbar region’s thermal signatures in the CLBP cases.

In a continued effort to deepen our understanding of CLBP and its manifestations, our study further explored the relationship between thermographic readings of the lumbar area and the presence of morning stiffness in the spine—a common symptom reported by individuals with CLBP. This particular analysis was predicated on the hypothesis that morning stiffness, as a clinical manifestation of CLBP, might be associated with specific thermal patterns detectable through IRT. Therefore, we meticulously examined the thermographic data collected from seven designated points (P1 to P7) across the lumbar region of participants, aiming to identify any significant thermal anomalies correlated with reports of morning stiffness. The outcomes of this investigation were systematically organized and are presented in Table 4 for a comprehensive review and analysis.

Upon analyzing the thermographic readings about the reported presence or absence of morning spinal stiffness among the study participants, we found that there were no statistically significant differences in the temperature values. This result indicates that, within the scope of our investigation, the occurrence of morning stiffness in individuals with CLBP did not appear to be associated with distinct thermal signatures in the lumbar area as measured through IRT.

In our investigation into CLBP, we explored whether thermographic readings correlate with recent acute low back pain “episodes” or sciatica incidents. The premise was that such acute episodes might influence the lumbar region’s thermal profile, indicating physiological or pathological changes. We thoroughly analyzed thermographic data from seven key points (P1 to P7) across the lumbar area of participants who reported a low back pain “crisis” or sciatica in the last two months, as detailed in Table 5. Our goal was to uncover any significant thermographic differences linked to these acute conditions, which could hint at underlying inflammatory or neurogenic processes. However, our examination revealed no statistically significant differences in thermographic values between those who had experienced recent acute episodes and those who had not. This finding suggests that, within the scope of our study, recent acute low back pain or sciatica episodes did not significantly lead to changes in the thermographic profiles of individuals with CLBP. This challenges the utility of IRT for identifying or distinguishing the thermal patterns associated with such acute episodes in CLBP cases.

In our study examining the link between clinical/physiological factors and thermographic profiles in CLBP sufferers, we used Spearman’s correlation coefficient, suited to non-parametric data, to assess the relationship strength and direction between various variables. These variables, including the average low back pain intensity over the last seven days, BMI, Schober test results, fatigue ratings, hand grip strength, and algometry scores, were analyzed against thermographic data from seven lumbar points (P1 to P7). Our aim was to identify any associations that could bring light to the physiological aspects of pain. Despite a comprehensive analysis, as shown in Table 6, we found no significant links between the intensity of reported low back pain and the thermographic patterns, indicating that within our research framework, pain intensity did not markedly influence the thermal profiles detected by IRT.

In the correlation between BMI, the Schober test, and the thermographic values in each position (P1 to P7) (see Table 7), statistically significant correlations were observed. Specifically, BMI was found to be negatively correlated with the thermographic values at P1 °C (r = −0.418 *), P2 °C (r = −0.508 **), and P3 °C (r = −0.361 *). This implies that individuals with higher BMI values tend to exhibit lower thermographic values at P1 °C, P2 °C, and P3 °C.

In examining the relationship between BMI, the results from the Schober test for lumbar flexibility, and the thermographic readings at specific positions (P1 to P7) on the lumbar region, our analysis revealed significant correlations, as detailed in Table 7. Notably, a negative correlation was observed between BMI and the thermographic readings at certain positions, specifically, at P1 °C (r = −0.418 *), P2 °C (r = −0.508 **), and P3 °C (r = −0.361 *). This indicates that as BMI increases, the thermographic readings at these positions tend to decrease, suggesting that higher BMI values are associated with lower temperatures at specific points in the lumbar area.

From the correlation between the Fatigue Assessment Scale items and the IRT values in each position (P1 to P7), in the pre-treatment (Table 8) in individuals with CLBP, there were statistically significant correlations; namely, the item “I feel physically exhausted” was positively correlated with the IRT value at P3 °C (r = −0.363 *), P4 °C (r = −0.384 *), and P7 °C (r = −0.593 **), and the item “I feel mentally exhausted” was positively correlated with the IRT value at P1 °C (r = −0.389 *) and P2 °C (r = −0.426 *). These results suggest that individuals who present higher values in the items on the Fatigue Assessment Scale tend to present higher IRT values in the mentioned positions.

A detailed examination of the connections between the average right-hand grip strength (measured in kilograms of force), the average left-hand grip strength (also measured in kilograms of force), and the thermographic measurements taken across seven distinct positions (designated P1 through P7), as thoroughly outlined in Table 9, demonstrated that there were no statistically significant relationships between these variables. This analysis was comprehensive, considering various factors and potential interactions, yet the data did not support any significant linkage between the manual grip strengths and the thermal readings across the specified positions.

The analysis investigating the relationship between lumbar algometry readings at the seven points and thermographic values across the same positions (P1 to P7), as shown in Table 10, did not reveal any statistically significant correlations.

The analysis of correlations between the lumbar algometry measurements (right side, left side, and central/spine) and thermographic readings at positions P1 to P7, as outlined in Table 11, found no statistically significant relationships.

## 4. Discussion

Advancements in camera technology have significantly influenced the adoption of IRT as a contactless method for measuring human skin temperature, commonly referred to as “tsk”. These technological improvements have not only enhanced the precision and functionality of thermal imaging cameras but have also made them more accessible due to reduced costs. Consequently, IRT stands out for its versatility, offering a non-invasive, contactless method for temperature measurements. This attribute is particularly beneficial in medical and research settings where minimizing physical contact is preferred. Systematic reviews have highlighted the effectiveness of infrared thermography in evaluating both inflammatory and degenerative joint diseases, underscoring its diagnostic value [25]. Additionally, the wireless nature of modern IRT systems facilitates seamless integration into various operational environments. Another notable feature of IRT is its ability to produce detailed thermograms, which visually allow temperature to be mapped across different areas of the skin, revealing both hot and cold spots. This capability is invaluable for diagnostic and monitoring purposes, as it provides insights into the physiological state of an individual, offering a window into underlying health conditions or the body’s response to environmental changes.

Although it was not possible with the software version used (FLIR Research Studio R&D software), in the future, it would be valuable to include representative thermographic maps and boxplots of temperature distribution at each point (P1–P7) to illustrate trends—even when not statistically significant—and to improve the visual interpretation.

However, methodological limitations inherent to the collection of IRT data may introduce confounding factors that affect temperature readings. Thus, it is crucial to control certain conditions effectively to reduce their impact on the measurement of skin temperature (tsk) using IRT. Achieving precise standardization is vital because protocol variations can influence the interpretation of results, image processing, or the consistency in selecting the region of interest [26]. Moreover, this lack of uniform protocols hampers the ability to compare data across studies, obstructing the development of a standardized dataset.

While various authors have set forth comprehensive guidelines for the use of IRT in clinical thermal imaging [27] and sports sciences [28], a gap has been noted in the provision of tailored recommendations for thermal imaging of patients with low back pain. In the context of clinical thermal imaging, it is imperative to maintain consistent environmental conditions, such as humidity and ambient temperature, to ensure the normalization of images. Additionally, securing the camera on a tripod is recommended to stabilize the imaging process. In our study, thermographic images were captured by a single examiner in an environment where humidity levels were maintained between 30% and 58%, and the temperature was kept within the range of 23 to 29 °C, with the camera positioned 0.8 m away from the subject. Given these specifics, we suggest adjustments to the existing criteria for conducting thermal imaging tests, aiming to enhance their applicability and precision, particularly for CLBP patients.

Protocols have been established for systematic reviews and meta-analyses assessing the diagnostic accuracy of infrared thermography in musculoskeletal injuries [24]. The protocol titled “thermographic imaging in sports and exercise medicine” (TISEM), devised by Moreira et al. in 2017, aimed to enhance the use of IRT by outlining a comprehensive approach for data collection and analysis [28]. The process includes gathering essential personal data from participants, issuing precise instructions, documenting the ambient temperature and humidity where the imaging takes place, and detailing the technical specifications of the camera, such as brand, model, accuracy, and emissivity settings. Additionally, it mandates an acclimation period, notes the exact timing of image captures, and requires the camera to be aligned perpendicularly to the targeted area. Despite these rigorous guidelines, certain uncontrollable factors remain, even with prior verbal confirmation from subjects. These include the consumption of alcohol, smoking, caffeine intake, eating substantial meals, and the application of ointments or cosmetics within four hours before the thermal imaging. Moreover, participants are advised against the use of anti-inflammatory medications, painkillers, contraceptives, and anesthetic agents for 48 h before undergoing the procedure.

Herein, when the participants reported specific conditions, these were documented, while some conditions were unavoidable and, thus, not grounds for exclusion. Instead, the participants were advised to refrain from consuming alcohol and caffeine, smoking, eating large meals, using ointments and cosmetics, and showering within four hours before their assessments. However, it is important to note that the TISEM guidelines were originally designed for IRT applications in sports and exercise medicine, not specifically for the scenarios being studied here. In this study, elderly patients were analyzed, many of whom presented a degenerative context, which contrasts with the population for whom these guidelines were originally developed.

In the analysis of thermographic data comparing the lower back temperatures of elderly individuals with CLBP against conventional benchmarks for this condition, no statistically significant differences were observed. However, several factors could account for the absence of statistical significance in our findings. Firstly, the limited size of our sample may have played a role. Additionally, significant variations in humidity were noted across different days of imaging, attributed to the data collection occurring in a room within thermal baths, leading to inconsistent humidity levels. Despite efforts to control environmental variables, the relatively broad variation in room temperature (23–29 °C) and humidity (30–58%) may have introduced inconsistencies in thermal readings. Future studies should aim to maintain temperature within ±1 °C and humidity below ±5% to ensure higher reliability. The choice to exclusively include elderly participants might also have limited our ability to detect notable temperature variations. Although the study focused on elderly individuals (66–88 years old), the findings cannot be extrapolated to younger populations. Including age-matched healthy controls in future work would provide clearer clinical relevance and benchmarking for interpretation. The absence of statistically significant results must be interpreted with caution. These null findings may reflect limitations such as small sample size, measurement variability, or unmeasured confounders, rather than a definitive lack of thermal differences. A previous study [29] in 2008 showed that older individuals tend to have lower temperature dissipation following physical activity in thermographic analysis compared to younger people. We speculate that this could be due to older adults having less hydrated tissues and greater adipose infiltration, which could affect the magnitude of temperature variation. This theory is bolstered by our findings that individuals with a higher BMI, likely to possess more adipose tissue, showed lower temperatures. The observed correlations between fatigue/BMI and lumbar temperature merit further exploration. For example, reduced muscle mass, altered posture due to fatigue, or hydration status could explain thermal variations in elderly individuals.

Regarding correlations, it was found that individuals scoring higher on the Fatigue Assessment Scale displayed elevated IRT readings at specific sites. This observation is hypothesized to stem from possible changes in spinal alignment in more fatigued individuals, who may tend to spinal flexion. This could increase the load on the lumbar region’s muscular structures, leading to a rise in temperature.

Previous studies have reported significant thermal differences in CLBP populations. However, our findings suggest that factors such as aging, comorbidities, and methodological variations could explain the discrepancy.

Given the relatively small sample size (*n* = 31), the statistical power may be limited, potentially affecting the ability to detect meaningful differences. Additionally, to enhance the consistency of thermographic measurements, future studies should aim to maintain the room temperature within ±1 °C and humidity variation below ±5%. Future research could benefit from higher-resolution thermographic cameras for improved spatial and thermal sensitivity. The FLIR i7 camera (140 × 140 resolution) used in this study, although accessible, has limited spatial resolution and thermal sensitivity compared to higher-end models. This may have constrained the precision of temperature detection.

## 5. Conclusions

This study aimed to explore the effectiveness of using IRT for the evaluation of CLBP in the elderly. The findings indicated a lack of correlation between the clinical manifestations of CLBP and the thermal patterns detected in thermographic images. More specifically, no correlations were found between lumbar thermographic profiles and demographic or clinical factors such as gender, age, pain characteristics, morning stiffness, and acute episodes in elderly individuals with CLBP.

This lack of correlation could be attributed to several factors, including an underappreciation of the physiological variability in skin temperatures compared to the specific anatomical sites of nerve damage, as well as the lack of tailored guidelines for thermal image acquisition in CLBP sufferers. This gap makes it challenging to align and compare data with findings from other research efforts.

Nonetheless, an interesting observation emerged from the study: individuals reporting lower fatigue levels and those with a higher BMI were generally associated with cooler thermal readings on the lumbar region’s thermographic maps.

Future research could explore the integration of artificial intelligence (AI) with infrared thermography, leveraging machine learning algorithms to enhance sensitivity, specificity, and pattern recognition in CLBP assessments. Additionally, a larger cohort will provide more robust findings. This combined approach has shown potential in enriching clinical assessments and could serve as a valuable complement to traditional diagnostic methods.

## Figures and Tables

**Figure 1 diagnostics-15-01317-f001:**
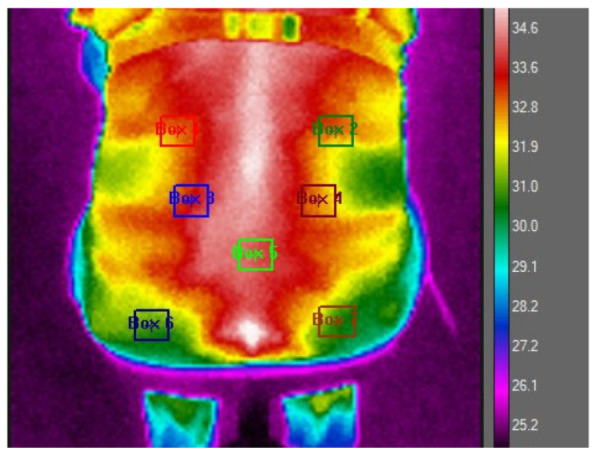
Thermal image illustrating the predefined regions of interest (ROIs) on the back.

**Table 1 diagnostics-15-01317-t001:** Comparison of the thermography values in each position (P1 to P7) between sexes.

Gender								
	Man			Woman				
	*n*	Mean	SD	*n*	Mean	SD	Dif.	*p*-Value
ROI								
P1_°C	9	32.53	1.94	21	32.59	1.39	−0.06	0.928
P2_°C	9	32.49	1.80	21	32.58	1.55	−0.10	0.874
P3_°C	9	33.07	1.49	21	33.32	1.42	−0.25	0.803
P4_°C	9	32.98	1.46	21	33.21	1.38	−0.22	0.964
P5_°C	9	33.95	1.07	21	33.65	1.24	0.30	0.483
P6_°C	9	32.12	1.25	21	31.82	1.68	0.30	0.402
P7_°C	9	31.95	1.56	21	31.59	1.60	0.36	0.415

(*p* = Mann–Whitney test; SD = standard deviation).

**Table 2 diagnostics-15-01317-t002:** Comparison of the thermography values in each position (P1 to P7) between age groups.

	64–75 Years Old			>75 Years Old				
	*n*	Mean	SD	*n*	Mean	SD	Dif.	*p*-Value
ROI								
P1_°C	18	32.56	1.55	12	32.58	1.59	−0.01	0.783
P2_°C	18	32.53	1.56	12	32.59	1.73	−0.05	0.611
P3_°C	18	33.07	1.57	12	33.52	1.18	−0.45	0.341
P4_°C	18	33.05	1.45	12	33.27	1.33	−0.21	0.582
P5_°C	18	33.61	1.40	12	33.93	0.76	−0.32	0.310
P6_°C	18	32.35	1.62	12	31.26	1.23	1.09	0.083
P7_°C	18	31.98	1.64	12	31.27	1.41	0.71	0.310

(*p* = Mann–Whitney test; SD = standard deviation).

**Table 3 diagnostics-15-01317-t003:** Comparison of the thermography values in each position (P1 to P7) between pain types.

	Lumbar Pain			Pain Radiating to the Lower Limbs				
	*n*	Mean	SD	*n*	Mean	SD	Dif.	*p*-Value
ROI								
P1_°C	9	32.08	2.06	21	32.78	1.25	0.70	0.342
P2_°C	9	31.96	2.18	21	32.81	1.26	0.85	0.455
P3_°C	9	32.94	1.77	21	33.38	1.27	0.44	0.769
P4_°C	9	32.92	1.75	21	33.24	1.23	0.32	0.946
P5_°C	9	33.61	1.40	21	33.80	1.11	0.19	0.910
P6_°C	9	31.90	1.12	21	31.92	1.73	0.01	0.946
P7_°C	9	31.47	1.19	21	31.80	1.72	0.32	0.684

(*p* = Mann–Whitney test; SD = standard deviation).

**Table 4 diagnostics-15-01317-t004:** Comparison of the IRT values in each position (P1 to P7) between morning stiffness levels.

	I Do Not Feel It, or It Is Less than 30 min			I Feel It, and It Is More than 30 min				
	*n*	Mean	SD	*n*	Mean	SD	Dif.	*p*-Value
ROI								
P1_°C	24	32.47	1.59	6	32.98	1.37	0.51	0.422
P2_°C	24	32.45	1.63	6	33.00	1.53	0.56	0.325
P3_°C	24	33.27	1.46	6	33.16	1.37	−0.11	0.917
P4_°C	24	33.18	1.41	6	32.97	1.38	−0.22	0.795
P5_°C	24	33.84	1.20	6	33.35	1.10	−0.49	0.568
P6_°C	24	32.17	1.54	6	30.89	1.22	−1.28	0.070
P7_°C	24	31.92	1.58	6	30.82	1.29	−1.10	0.102

(*p* = Mann–Whitney test; SD = standard deviation).

**Table 5 diagnostics-15-01317-t005:** Comparison of the IRT values in each position (P1 to P7) between episodes and non-episodes of low back pain or sciatica.

	In the Past Two Months, Have You Experienced Any “Crises” of Low Back Pain or Sciatica?							
	No			Yes				
	*n*	Mean	SD	*n*	Mean	SD	Dif.	*p*-Value
ROI								
P1_°C	16	32.30	1.46	14	32.87	1.62	0.57	0.383
P2_°C	16	32.36	1.59	14	32.78	1.64	0.41	0.647
P3_°C	16	32.86	1.28	14	33.69	1.50	0.82	0.114
P4_°C	16	32.91	1.29	14	33.41	1.49	0.50	0.442
P5_°C	16	33.34	1.02	14	34.20	1.22	0.87	0.061
P6_°C	16	31.63	1.17	14	32.23	1.89	0.59	0.339
P7_°C	16	31.28	1.16	14	32.18	1.86	0.89	0.212

(*p* = Mann–Whitney test; SD = standard deviation).

**Table 6 diagnostics-15-01317-t006:** Correlation between the average value of low back pain in the last 7 days and the IRT values in each position (P1 to P7).

	How Do You Rate the Average Value of Your Low Back Pain over the Last 7 Days?
ROI	
P1_°C	0.257
P2_°C	0.250
P3_°C	0.131
P4_°C	0.118
P5_°C	0.158
P6_°C	−0.027
P7_°C	0.114

(*p* = Spearman’s correlation coefficient).

**Table 7 diagnostics-15-01317-t007:** Correlation between BMI, the Schober test, and the IRT values in each position (P1 to P7).

	BMI	Schober Test
IRT		
P1_°C	−0.418 *	−0.020
P2_°C	−0.508 **	−0.040
P3_°C	−0.361 *	−0.078
P4_°C	−0.227	−0.153
P5_°C	−0.340	−0.092
P6_°C	−0.015	−0.100
P7_°C	0.094	−0.173

* The correlation is significant at the 0.05 level. ** The correlation is significant at the 0.01 level.

**Table 8 diagnostics-15-01317-t008:** Correlation between items from the Fatigue Assessment Scale and the IRT values in each position (P1 to P7).

	Fatigue Assessment Scale				
	Fatigue Bothers Me	I Get Tired Very Quickly	I Feel Physically Exhausted	I Feel Like I Don’t Want to Do Anything	I Feel Mentally Exhausted
ROI					
P1_°C	0.062	0.188	0.244	0.224	0.389 *
P2_°C	−0.006	0.241	0.234	0.228	0.426 *
P3_°C	−0.042	0.12	0.363 *	0.118	0.358
P4_°C	−0.085	0.13	0.384 *	0.087	0.348
P5_°C	0.015	0.169	0.335	0.137	0.211
P6_°C	−0.15	−0.099	0.29	0.003	0.223
P7_°C	0.038	0.217	0.593 **	0.082	0.189

* The correlation is significant at the 0.05 level. ** The correlation is significant at the 0.01 level.

**Table 9 diagnostics-15-01317-t009:** Correlation between the manual grip R (KgF), manual grip LE (kgF), and IRT values in each position (P1 to P7).

	Hand Grip (kgF)	
	R, Mean	L, Mean
ROI		
P1_°C	−0.046	−0.012
P2_°C	−0.070	−0.038
P3_°C	−0.113	−0.097
P4_°C	−0.136	−0.049
P5_°C	0.028	0.016
P6_°C	−0.059	0.048
P7_°C	−0.048	0.068

**Table 10 diagnostics-15-01317-t010:** Correlation between the lumbar algometry and IRT values in each position (P1 to P7).

	Lumbar Algometry (kgF)						
	ALG_1	ALG_2	ALG_3	ALG_4	ALG_5	ALG_6	ALG_7
ROI							
P1_°C	0.090	−0.009	−0.035	−0.067	−0.113	0.031	−0.005
P2_°C	0.135	0.089	0.043	0.011	−0.050	0.060	0.087
P3_°C	−0.069	−0.026	−0.053	−0.114	−0.057	0.159	−0.028
P4_°C	−0.273	−0.185	−0.136	−0.261	−0.156	−0.058	−0.243
P5_°C	−0.082	−0.031	−0.026	−0.041	−0.026	0.258	−0.022
P6_°C	−0.090	−0.024	0.040	−0.101	0.082	0.116	−0.045
P7_°C	−0.104	−0.033	0.036	−0.075	0.144	0.141	−0.047

**Table 11 diagnostics-15-01317-t011:** Correlation between the lumbar algometry (right, left, and central) and IRT values in each position (P1 to P7).

	Lumbar Algometry (kgF)		
	Right	Left	Central
ROI			
P1_°C	0.037	−0.066	0.031
P2_°C	0.093	0.028	0.060
P3_°C	0.018	−0.070	0.159
P4_°C	−0.174	−0.249	−0.058
P5_°C	0.044	−0.042	0.258
P6_°C	0.014	−0.057	0.116
P7_°C	−0.002	−0.046	0.141

## Data Availability

The data presented in this study are available from the corresponding author upon reasonable request, due to ethical and privacy restrictions.

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
