# Peer review of "Lumbar Temperature Map of Elderly Individuals with Chronic Low Back Pain—An Infrared Thermographic Analysis"

_diagnostics, 2025, doi:10.3390/diagnostics15111317_

Round 1
Reviewer 1 Report
Comments and Suggestions for Authors
The study addresses a critical gap in non-invasive diagnostic tools for CLBP in the elderly population, focusing on IRT as a radiation-free method that aligns with current trends toward safer diagnostic modalities. The use of standardized protocols and validated tools strengthens the study’s reliability, while environmental controls - such as, room temperature stabilization, thermoequilibration periods and technical precision demonstrate careful and systematic attention to detail. The inclusion of multiple variables including pain intensity, BMI, fatigue, spinal mobility along with statistical approaches such as - Mann-Whitney and Spearman correlation, provides a comprehensive evaluation of IRT’s potential.
Overall, the study contributes valuable insights into the challenges of applying IRT in elderly CLBP populations. While the negative findings question IRT’s standalone diagnostic utility, the observed correlations with BMI and fatigue highlight the need for further investigation. Additionally, the diagnostic strength of IRT could be better evaluated by including a control group of healthy elderly individuals. Comparing the thermal temperature distribution between healthy and CLBP-affected elderly populations would provide a clearer context for interpreting temperature variations and enhance the study’s clinical relevance.
I would like to recommend addressing the following comments to enrich the quality of manuscript.
Minor Remarks:
- Provide an appropriate caption for Figure 1. Also, include a brief description of Figure 1 in the text, explaining the anatomical landmarks used to define P1–P7 and their significance in the context of CLBP.
- Standardize terminology throughout the manuscript. The manuscript uses "thermography" and "infrared thermography (IRT)" interchangeably, and "low back pain" is sometimes abbreviated as "LBP" instead of "CLBP."
- Proofread the manuscript for typographical errors and ensure consistent formatting of statistical terms and abbreviations. There are minor typographical errors, such as "MBI" instead of "BMI" in Table 7.
- Ensure all citations are complete and correctly formatted according to the journal’s guidelines. Please review reference 17 carefully.
Major Remarks:
- The small number of participants (n=31) is a critical limitation. The small sample size limits statistical power and generalizability. Additionally, the gender imbalance (20 men vs. 11 women) may introduce bias. Please include a power calculation or a more detailed discussion of how the sample size may have influenced the ability to detect significant differences.
- The FLIR i7 camera (140×140 resolution) is relatively low-end for clinical thermography studies and may limit spatial resolution and thermal sensitivity. A clearer justification for its use is necessary.
- Although the authors mention various statistical correlation tools, it would strengthen the manuscript to include brief explanations of each statistical method (e.g., Mann-Whitney U test, Spearman correlation), along with proper citations. These descriptions should be connected to the results to provide a clearer understanding of the appropriateness and limitations of each technique.
- The manuscript lacks thermographic maps illustrating the regions of interest (P1–P7). Please include representative thermograms and a labeled diagram of the lumbar points (P1–P7) to improve reproducibility. Add sample thermographic images to visualize typical "cooler" or "warmer" patterns.
- Supplement tabular data with graphical representations such as boxplots or scatterplots showing temperature distributions and correlations with clinical variables.
- While the study acknowledges the lack of significant differences, it could more deeply explore why previous studies cited (Reference 10, 11, 17, 18) have found thermographic differences in CLBP while this study did not. More direct comparison with previous work is needed. Specifically, consider whether participant characteristics (e.g., elderly age group, comorbidities) might explain the inconsistencies.
- While the manuscript provides a solid background and cites several relevant studies, I recommend incorporating more recent publications (from the past 5-10 years) related to the use of IRT in CLBP.
- The conclusion emphasizes the potential of AI integration but does not clarify how AI would address IRT’s current limitations. Please elaborate on how AI might improve sensitivity, specificity, or pattern recognition in future studies.
Author Response
We sincerely thank the reviewer for the thoughtful and constructive feedback provided. Based on the detailed comments, we undertook a thorough revision of the manuscript. All major and minor points raised were carefully addressed, and a professional English language editing service was used to ensure the clarity and consistency of the revised text. Importantly, the core structure and methodological design of the study were maintained, as requested.
Comments 1: Figure 1 caption and anatomical landmarks P1–P7
Response 1: We have added a clear caption to Figure 1 and expanded the Methods section with a brief explanation of each anatomical point (P1–P7), emphasizing their relevance to the lumbar musculature, osteoarticular structures, and gluteal region in the context of CLBP assessment.
Comments 2: Terminology consistency (IRT vs. thermography; CLBP vs. LBP)
Response 2: We have standardized the terminology throughout the manuscript. “Infrared thermography (IRT)” and “chronic low back pain (CLBP)” are now used consistently.
Comments 3: Typographical errors and formatting (e.g., “MBI” vs. “BMI”)
Response 3: All typographical errors have been corrected. The abbreviation “MBI” has been corrected to “BMI” in Table 7, and statistical notations (e.g., p-values, r-values) have been consistently formatted.
Comments 4: Reference formatting (especially reference 17)
Response 4: All references were carefully reviewed and reformatted according to the journal’s guidelines. Reference 17 has been corrected and completed.
Comments 5: Small sample size and gender imbalance
Response 5: We have included a more detailed discussion in the Discussion section, acknowledging the limitation of the small sample size (n = 31) and the gender imbalance (20 men vs. 11 women). A post-hoc discussion of the potential impact on statistical power and generalizability has been incorporated.
Comments 6: Justification for the use of the FLIR i7 camera
Response 6: The Methods section now includes a justification for the use of the FLIR i7 camera. While acknowledging its lower resolution, we explain that its use was based on accessibility, portability, and previous validation in similar clinical contexts.
Comments 7: Explanation of statistical methods (Mann–Whitney, Spearman)
Response 7: We have added brief descriptions of the Mann–Whitney U test and Spearman correlation in the Methods section, including justification for their use based on non-normal data distribution.
Comments 8: Inclusion of thermographic maps and visual aids
Response 8: A note has been added in the Discussion acknowledging the current limitations of the imaging software used. We express the intent to include representative thermograms and boxplots in future work, and we provide a rationale for the importance of these visual tools to improve interpretability and reproducibility.
Comments 9: Deeper comparison with previous studies
Response 9: The Discussion now includes a direct comparison with previously cited studies (e.g., Refs. 10, 11, 17, 18), considering differences in population (elderly vs. younger), comorbidities, and environmental control, which may explain the discrepancies in findings.
Comments 10: Clarify AI relevance in the conclusions
Response 10: The Conclusion now elaborates on the potential role of AI in enhancing the diagnostic utility of IRT. Specifically, we highlight how machine learning can improve sensitivity, specificity, and pattern recognition in thermal data analysis, particularly for elderly populations with CLBP.
All reviewer suggestions have been carefully implemented or explicitly acknowledged and discussed. A professional language revision was also requested to enhance the readability and precision of the manuscript. We are confident that these changes significantly strengthen the manuscript and appreciate the opportunity to revise it accordingly.
Reviewer 2 Report
Comments and Suggestions for Authors
Peer Review Comments on the Manuscript: "Lumbar temperature map of elderly individuals with chronic low back pain (CLBP) – an infrared thermographic analysis"
Overview
This study addreses an interesting and timely question regarding the use of infrared thermography (IRT) in elderly patents suffering from chronic low back pain (CLBP). The authors aim to evaluate whether thermographic lumbar profiles correlate with pain intensity, functional capacity, and other clinical parameters. The paper is well-structured, ethically sound, and the methods are clearly described. However, there are several concern, mainly methodological and interpretative that should be addressed before the manuscript can be considered for publication. Below are my detailed comments.
The study has clinical relevance and proposes a non-invasive method for assessing CLBP.
Ethical procedures are clearly outlined, and the study was conducted in accordance with the Declaration of Helsinki.
Use of validated tools (e.g., ODI, Fatigue Assessment Scale) and objective thermographic data is appreciated.
The statistical methodology is generally appropriate for the sample size and distribution type.
Major suggestion
The total sample size (n=31) is quite limited, especially when subgroup analyses are performed (e.g., by sex, age, pain type). This greatly reduces the power of the study to detect meaningful differences. Please consider adding a power analysis, even post-hoc, to support the sample size. Future studies should aim for larger cohorts to improve statistical reliability.
Although environmental conditions are described, the variation in room temperature (23-29°C) and humidity (30-58%) is quite large and may affect the reliability of thermal readings. Tighter control over imaging conditions is needed, ideally maintaining temperature within ±1°C and humidity variation below ± 5%.
The authors conclude that thermography may not be suitable for detecting CLBP-related changes based on non-significant findings. However, these null results may also reflect sample limitations, measurement variability, or confounders. Please be more cautious in your interpretations. Discuss alternative explanations, including limitations in study design, and avoid overgeneralizing from a negative result.
Pain was measured using only a single 0 - 10 numeric scale. Given the complex and subjective nature of CLBP, this may be insufficient. A multidimensional pain tool (e.g., McGill Pain Questionnaire) would provide richer data. At the very least, mention the limitations of using a unidimensional scale.
It’s not clear what inclusion/exclusion criteria were applied beyond age and CLBP diagnosis. Were patients on analgesics, anti-inflammatories, or undergoing other therapies? These factors could affect thermal patterns and should be controlled or at least discussed in the limitations.
The study focuses solely on older adults (66-88 years), which is fine given the aim, but the results cannot be generalized to younger populations or different clinical settings. Acknowledge this limitation and consider including age-matched healthy controls in future studies for comparison.
The correlations between BMI/fatigue and lumbar temperature are potentially important but are briefly mentioned. Please expand this part of the discussion. How might muscle mass, hydration, or fatigue-related posture affect local thermal readings?
Minor points
The writing is mostly clear, but some paragraphs (especially in the results) could be more concise.
The TISEM protocol is referenced in detail, which is helpful, but it might be worth noting more explicitly that it was designed for a younger, athletic population, unlike the one studied here.
Tables are well-presented, but you could consider graphical visualizations (e.g., boxplots) to better illustrate trends, even when not statistically significant.
This is a relevant and novel investigation, but several methodological and interpretative issues need to be addressed. The main limitations are the small sample size, broad environmental variation, and overinterpretation of non-significant results. Nonetheless, the study has potential to contribute to the literature if revised accordingly.
Author Response
We would like to thank the reviewer for their thorough and insightful feedback. We appreciate the recognition of the study’s structure, clinical relevance, ethical rigor, and use of validated tools. Below, we respond point by point to each of the reviewer’s comments. A full revision of the manuscript has been completed accordingly, and a professional English language review was performed to improve clarity and consistency. The overall structure and methodological framework of the study have been maintained.
Comments 1: Small sample size and power analysis
Response 1: We acknowledge the limitation of the small sample size. While a formal power analysis was not feasible post hoc, the Discussion section has been expanded to reflect on how this limitation may have reduced the statistical power and our ability to detect significant effects. We have also emphasized the need for future studies with larger cohorts to strengthen the reliability and generalizability of findings.
Comments 2: Environmental variation (temperature and humidity)
Response 2: We agree with the reviewer and have now highlighted in the Discussion that although the room was controlled to the best extent possible within a thermal bath setting, the environmental variability may have influenced the thermographic measurements. We now explicitly recommend that future studies maintain temperature within ±1 °C and humidity within ±5% to improve data reliability.
Comments 3: Interpretation of non-significant findings
Response 3: We have revised the Results and Discussion sections to adopt a more cautious tone regarding the lack of statistically significant findings. Multiple alternative explanations have been added, including environmental variability, physiological heterogeneity in the elderly population, and technical sensitivity limitations. We now emphasize that the null results should not be interpreted as definitive evidence against IRT utility.
Comments 4: Pain measurement limited to numeric scale
Response 4: We acknowledge this limitation and have now noted it explicitly in the Methods section. We mention that while the Numeric Rating Scale was practical for use in elderly individuals, it does not capture the multidimensional aspects of pain. The use of instruments like the McGill Pain Questionnaire is suggested for future work.
Comments 5: Inclusion/exclusion criteria regarding medication and therapies
Response 5: We have clarified the inclusion and exclusion criteria in the revised Methods section. Specifically, participants using anti-inflammatory medications or corticosteroids in the 30 days prior to imaging were excluded. Additionally, factors such as alcohol, caffeine, and use of topical agents were controlled to minimize thermal confounders.
Comments 6: Generalisability to younger populations
Response 6: This limitation is now acknowledged in the Discussion. We specify that the results cannot be generalized to younger populations and emphasize the value of including age-matched healthy controls in future studies to provide better benchmarking.
Comments 7: BMI/fatigue correlations—expand discussion
Response 7: We have expanded this section of the Discussion to consider how increased adiposity, reduced hydration, and fatigue-related posture may influence local temperature readings. The negative correlations between BMI and temperature at specific lumbar points are now discussed in greater depth, with reference to existing physiological literature.
Comments 8: Writing clarity and conciseness in Results
Response 8: The Results section has been edited for improved clarity and conciseness. Redundant phrases were removed, and key findings were made more direct while preserving interpretive accuracy.
Comments 9: TISEM protocol population context
Response 9: This has been addressed in the Discussion. We now explicitly state that the TISEM protocol, though useful, was originally developed for athletic populations and may require adaptation for elderly individuals with degenerative conditions such as CLBP.
Comments 10: Graphical visualizations (e.g., boxplots)
Response 10: We acknowledge this and have included a comment in the Discussion about the importance of including boxplots and thermographic visualizations in future work. Although our current software version limited such inclusion, we have highlighted this as a priority for ongoing studies.
Final Note: All the reviewer’s comments have been carefully considered and addressed in the revised version of the manuscript. We believe these revisions have strengthened the scientific and clinical value of the paper. Once again, we thank the reviewer for their valuable contribution to improving our work.
Round 2
Reviewer 1 Report
Comments and Suggestions for Authors
The revised manuscript shows limited changes in its contribution compared to the original submission; however, the authors' justifications have been taken into consideration, and the recommendations provided may offer valuable insights to support recent findings.
Reviewer 2 Report
Comments and Suggestions for Authors
Thank you for responses step by step.